# Duration, frequency, and time distortion: Which is the best predictor of problematic smartphone use in adolescents? A trace data study

**Laura Marciano**[ID]*, **Anne-Linda Camerini**[ID]

Institute of Public Health, Università della Svizzera italiana, Lugano, Switzerland

* laura.marciano@usi.ch

## Abstract

Problematic smartphone use (PSU) during adolescence has been associated with negative short- and long-term consequences for personal well-being and development. Valid and reliable predictors and indicators of PSU are urgently needed, and digital trace data can add valuable information beyond self-report data. The present study aimed to investigate whether trace data (duration and frequency of smartphone use), recorded via an app installed on participants' smartphone, are correlated with self-report data on smartphone use. Additionally, the present study aimed to explore which usage indicators, i.e., duration, frequency, and time distortion of smartphone use, better predict PSU levels cross-sectionally and longitudinally, one year later. Results from a sample of 84 adolescents showed that adolescents tend to rely on the frequency of smartphone use when reporting on the time they spent with the smartphone. Traced duration of smartphone use as well as time distortion, i.e., over-estimation, are significant predictors of PSU. Methodological issues and theoretical implications related to predictors and indicators of PSU are discussed.

## Introduction

The rise of digital media has sparked numerous studies on adolescents' use of these media, frequently focusing on the associations with developmental and well-being outcomes [1–7]. A large body of previous studies is concerned with the negative aspects of too much media use, especially adolescents' dependency on digital devices like smartphones. In fact, the smartphone has become an indispensable part of most adolescents' life, and its use is driven by the core motivations and needs of that age [8]. According to a recent review, one out of four adolescents shows symptoms of Problematic Smartphone Use (PSU) [5], which negatively affects social and emotional well-being [9] and academic outcomes [10]. However, until today, most of the studies focusing on PSU relied on self-report data making it hard to draw valid conclusions on which type of smartphone activity can be described as problematic [11]. Yet, technological advancements have enabled the use of digital trace data–defined as "records of activity undertaken through an online information system—thus, digital" [12] [p767] –in addition to or

**Data Availability Statement:** The study data and questionnaire are available here: https://osf.io/hwr2u

**Funding:** This research has received funding from the Swiss National Science Foundation under the Grant no. 10001C_175874. The funders had no role in study design, data collection and analysis, decision to publish, or preparation of the manuscript.

**Competing interests:** The authors have declared that no competing interests exist.

even as a substitute for self-report data. To date, few studies made use of trace data to investigate PSU, focusing on adult populations [for a review, see 13]. The incorporation of digital trace data in research with adolescents is still scarce [14–16]. Therefore, the present study wanted to fill the gap and investigated adolescents' PSU by considering both self-report and digital trace data of smartphone use at the cross-sectional and longitudinal level. This approach shed light on the behavioral patterns that lead to the development of PSU, thus informing educational interventions that promote a *smart* use of the smartphone in adolescence.

## Literature review

### Problematic smartphone use

Smartphones have become the preferred digital media device among adolescents. In the U.S., 95% of teens have access to a smartphone, through which they mostly pass time, connect with others, learn new things, and avoid face-to-face interactions [17]. In Western Europe, prevalence rates are similar. In Switzerland, 99% of 12 to 19-year-olds own a smartphone, on which they spend, on average, almost 4 hours during a weekday and more than 5 hours during a weekend day [13]. Early adolescents (12 to 13 years) spend more time playing online games, while they make less use of the smartphone for instant messaging, social media, online searches, and activities like listening to music and creating multimedia material. On the contrary, older adolescents use the smartphone more often for social media, photo and video editing, time management, e-mail, navigation, and vocal assistance [13]. In 2020, the amount of time spent on the device reached the highest increment in the last ten years, also due to the COVID-19 pandemic and associated social distancing and confinement measures [13], thus making it even more imperative to comprehend PSU objectively. The easiness of its portability, the ubiquitous accessibility, and the multitude of personalized Internet-based applications on the smartphone potentially facilitate the development of PSU. Real-time data showed that youth pick up their smartphone every five minutes, thus making smartphone use a deeply internalized, unconscious, and reinforced behavior [18]. As reported by Sohn and colleagues [5], one in four children and young people shows signs of PSU. According to the same review, PSU is consistently associated with higher levels of depression, anxiety, stress, sleep problems, and daily functional impairment, including poor academic achievement. Similar results have been found in other studies, where PSU was linked to internalizing symptoms, risk of cyberbullying behaviors, higher distractibility, and body image concerns, to name a few [10,19,20].

Although a standard agreement on what constitutes PSU does not yet exist, it has been compared to the similar and better-researched concept of Internet Addiction (IA). PSU and IA share similarities with the more general category of "behavioral addictions", introduced in 2010 by the DSM-5 Working Group after decades of debate [21,22]. Behavioral conducts carried out in an extreme way generate substantial problems for a person's well-being, independently of the nature of the specific activity. The repetitive and compulsive engagement in smartphone-related activities shares common characteristics with substance use disorders and gambling disorder (the first disorder included as a non-substance behavioral addiction in the DSM-5). In the case of PSU, these characteristics comprise cognitive salience, spending a lot of time using the smartphone, unsuccessful attempts to reduce smartphone use, positive mood when using the device and symptoms like irritability and distress when one is unable to reach it, difficulty in regulating the use, unsuccessful attempts to stop, and interpersonal problems (with family or friends, at work or school) due to excessive smartphone use [23–28]. PSU is also closely linked to "nomophobia", which refers to discomfort, anxiety, nervousness or anguish caused by being without a mobile phone [29]. To summarize, although diverse terminologies and measures exist [30], PSU has been generally defined akin a behavioral or non-

substance related addiction. However, it is important to consider that smartphones are linked to different (social) rewards, such as positive feedback and validation by peers [8]. This makes adolescents use the smartphone many times a day without necessarily reflecting an addictive behavior [31–33]. That said, self-report measures can only partially depict different features of PSU.

## Overcoming self-report biases

It is commonly acknowledged that self-report is subject to systematic biases, which include, among others, recall, estimation, and social desirability bias [34–36]. Recall bias, for example, is the result of cognitive burden and occurs when respondents use heuristic shortcuts to recall the duration and frequency of everyday behaviors [37]. Furthermore, problems with time estimation are frequent, especially among younger populations who still have to develop a sense of time and the ability to quantify the time they engaged in different activities and to report on it [38]. On the other hand, social desirability bias is a systematic bias across all age. It describes the tendency of respondents to inaccurately report on sensitive topics to present themselves in the best possible light [39]. Self-report biases are a recognized methodological problem in different domains, including health sciences [40], research on organisations [41], consumer marketing [42], and media studies [43]. Concerning smartphone use, a recent review and meta-analysis underlined that, although several self-report scales correlate with objectively logged data, the strength of the relationship is far from convincing [44]. To this regard, the inclusion of objective trace data would complement the assessment of PSU.

Thanks to technological advancements, automated recording tools, such as tracking applications [45], are now available to overcome these biases [for an introduction, see 46], and they represent an excellent potential in incorporating digital trace data in survey research. However, the use of tracking applications to assess smartphone use is still at the beginning [46–48], and researchers pursuing the collection and analysis of trace data face different challenges. In their overview paper on integrating survey and digital trace data, Stier and colleagues [47] highlighted three potential issues: Firstly, as with any study, the collection of trace data requires informed consent from individuals who may have objections about automated tracking of their digital media behaviors. Secondly, technical and methodological issues can cause problems in the collection and scientific analysis of digital trace data alone and in combination with survey data. Thirdly, researchers need to improve the conceptual and theoretical frameworks for dealing with such data's richness. A fourth challenge can be added to the list, namely, legal restrictions. For example, the European Union does not allow collecting shared digital media content with identifiable information about depicted persons without the active consent of all parties represented in this content (i.e., third-party consent) [49].

Despite these challenges in collecting digital trace data, researchers have started to trace smartphone use as an increasing phenomenon among younger populations across different societies [50–52]. Through smartphones, automated recording can happen in very different ways, for example, by tracking users *via* dedicated applications installed on the device [47] or *via* a software-modified smartphone to record the number and duration of calls [53]. For example, Mireku and colleagues [14] examined the similarity between self-reported mobile phone use data and objective mobile operator traffic data in 11–12 years old participants, focusing on calls and text messages. The number and duration of calls done by adolescents were also assessed by Inyang and colleagues [15] and Aydin and colleagues [16]. In addition, Reeves and colleagues [51,54] developed a technology based on a series of screenshots of the user's mobile device to create what the authors call a 'screenome' of digital media use.

## Investigating PSU using objective trace data

Since the smartphone's excessive use is an indicator of PSU [23], researchers have started to investigate the frequency and duration of smartphone use by incorporating objective trace data in their statistical analyses [55]. When it comes to the exploration of behavioral patterns of PSU, Lin and colleagues proposed diagnostic criteria for what the authors called "smartphone addiction" based on digital trace data and clinical interviews in a sample of adult participants [50]. Additionally, in another study [56], they found that the presence of underestimation (i.e., the discrepancy between traced duration of device use and self-reported use) and the traced frequency of smartphone use were better predictors of psychiatrists' rating of smartphone addiction than the traced duration of device use. This result stands in partial contrast to a previous research conducted by Ko and colleagues [57], who found that self-report excessive time spent online (i.e. duration of use) was a risk factor of problematic Internet use as well as PSU. Furthermore, Tossell and colleagues [58] compared smartphone use indicators among young adults grouped into addicted and non-addicted smartphone users according to their self-assessment. They found that the self-declared addicts spent twice the time on their phones and started interacting with applications (especially those meant for social interactions) twice as often as the non-addicts. Likewise, a study by Noë and colleagues [59] revealed that user-interface interactions, especially if they involve lifestyle and social media applications, were related to higher PSU levels in a sample of young adults. To the best of our knowledge, and according to a recent review summarizing objective measures used to assess PSU [55], no study to date has combined self-report and digital trace data to predict PSU in mid adolescents cross-sectionally, let alone longitudinally.

## Study aim

To fill this gap, the present study aimed to identify valid and reliable indicators of PSU in adolescence over time by using self-report and objective trace data on smartphone use. From a methodological point of view, the study wanted to explore to what extent self-report data are an accurate estimate of objective digital trace data. From a theoretical point of view, the study aimed to investigate which indicators among traced duration of smartphone use, traced frequency of use, and time distortion (i.e., the discrepancy between self-report and traced use), best predict PSU cross-sectionally and longitudinally, one year later. Based on the findings from previous studies with adult populations, we derived the following hypotheses:

*H1*: *Longer duration of traced smartphone use is related to higher levels of PSU.*

*H2*: *Time distortion of smartphone use is related to higher levels of PSU.*

*H3*: *Higher frequency of traced smartphone use is related to higher levels of PSU.*

Additionally, we added the following two research questions:

*RQ1*: *How much does self-report smartphone use correlate with traced duration of smartphone use?*

*RQ2*: *Is there a difference in the correlation between self-report duration and traced duration when comparing smartphone use for weekdays and weekend days?*

## Methodology

The present study is part of the longitudinal MEDIATICINO panel study (www.mediaticino. usi.ch), focusing on digital media use and youth well-being. Since 2014, the study has followed

a cohort of approximately 1'400 adolescents born in 2004/05 and residing in Canton Ticino, Italian-speaking Switzerland.

## Data collection

**Survey data.** The larger panel study relies on an annual self-administered paper-and-pencil questionnaire to collect information about adolescents' use of digital media (including duration and type of use), physical, psychological, and social well-being, as well as on different aspects of the parent-child relationship. Each year, survey data are collected in collaboration with schools, who distribute the questionnaire using a Unique Identifier (U-ID) and the associated student name, to which only school staff has access. The U-ID is used to match different waves and assure participants' anonymity during data processing and analysis by the research team. Further information on the data collection procedure can be found elsewhere [60].

**Digital trace data.** In 2018 and 2019, the panel study was extended by introducing an automated way to collect data on smartphone use, i.e., by recording data of adolescents' smartphone use *via* a dedicated application installed on their device. All the panel study participants were invited to participate through a letter forwarded by the collaborating schools to students' families. Participants who provided parental consent received further information on how to download the application and register for the study. The application, called *Ethica*, was specially developed for public health research purposes for Android and iOS operating systems (ethicadata.com). *Ethica* automatically collects trace data, such as screen time and application usage. To match trace data with self-report questionnaire data, participants in the *Ethica* study received a generated login e-mail address matched to their U-ID. Schools distributed the instruction material with information on how to download *Ethica* and register for the study within the application by using the generated login e-mail address and a personal password. This procedure allowed the combination of digital trace data and questionnaire data while guaranteeing students' anonymity. Students provided their active consent directly in the *Ethica* application upon enrolment.

**Ethical considerations.** The Cantonal education administration of Ticino approved the annual panel study based on self-administered questionnaires. The embedded *Ethica* study received additional approvals from the Ethics Committee of the Università della Svizzera italiana and from the Cantonal Data Protection Officer of Ticino.

## Sample

In 2018, the larger panel included 1419 students, of which 1374 (96.8%) completed the paper-and-pencil questionnaire at school. For 264 (18.6%) students, parents provided informed consent to invite their children to the *Ethica* study. Despite parental consent, 169 (64%) students did not download the *Ethica* application. The remaining 95 students (6.7% of the initial sample) eventually participated in the *Ethica* study in 2018. Compared to the students who did not join the *Ethica* study, included participants did not differ in gender (p = .205), perceived socio-economic status (p = .229), or self-reported daily smartphone use (p = .114) [61]. Due to technical problems and some participant dropouts, reliable trace data were available for 89 participants. When combining digital trace data and self-report data, we eliminated other five cases with missing values on self-report data. The analytical sample was composed of 84 participants with complete and matched data for T1 ($M_{age}$ = 13.56, $SD_{age}$ = .52, 46.4% males), of which 80 participants also had complete survey data for T2.

## Measures

**Perceived duration of smartphone use.** Self-report smartphone use was measured in the annual survey with two questions: "How much time do you usually use the smartphone on a

typical school day/weekend day?". For each question, students estimated their daily smartphone use by choosing one option on a scale with nine-time interval: 0 "never", 1 "up to 0.5 hours", 2 "between 0.5 and 1 hour", 3 "between 1 and 1.5 hours", 4 "between 1.5 and 2 hours", 5 "between 2 and 3 hours", 6, "between 3 and 4 hours", 7 "between 4 and 5 hours", and 8 "5 or more hours". To allow the comparison with trace data, results were converted into hours by using the midpoint for each category of the original interval scale: (0 = 0) (1 = 0.25) (2 = 0.75) (3 = 1.25) (4 = 1.75) (5 = 2.5) (6 = 3.5) (7 = 4.5) (8 = 5.5). For the highest interval ("5 or more hours"), we used 5.5 hours as a proxy of time spent with the smartphone [62]. The weighted mean between a typical school day and a typical weekend day was used as an approximate measure of smartphone use in terms of hours *per* day [(estimation of a typical weekday*5+ estimation of a typical weekend day*2)/7] (M = 1.88, SD = 1.41, r between weekday and weekend day use = .90, p < .001).

**Trace data for the duration of smartphone use.** The traced duration was divided into hours on weekdays and weekend days. All weekdays were averaged to obtain an aggregate measure for typical weekday use, and all weekend days were averaged to obtain an aggregate measure for typical weekend day use. Once again, the weighted mean between a typical weekday and a typical weekend day was used as an approximate measure of smartphone use in terms of hours *per* day [(traced duration for a typical weekday*5+traced duration for a typical weekend day*2)/7] (M = 1.89, SD = 1.52, r between weekday and weekend day use = .855, p < .001). Missing data were handled with Hidden Markov Models [for more details, see 63].

**Trace data for frequency of smartphone use.** The frequency of smartphone use, i.e., checking behavior, was assessed by automatically counting how many times participants activated their smartphone screen during a weekday and a weekend day. Applying the abovementioned formula, we obtained an average measure of checking behavior *per* day weighted for weekdays and weekend days [(traced frequency for a typical weekday*5+traced frequency for a typical weekend day*2)/7] (M = 57.46, SD = 38.07, r between weekday and weekend day use = .888, p < .001)

**Problematic smartphone use.** Perceived PSU was assessed with the short version of the Smartphone Addiction Scale for adolescents (SAS-SV; 62) at both T1 in spring 2018 and T2 in spring 2019. The scale consists of ten items measured on a scale from 1 "strongly disagree" to 6 "strongly agree". All items were averaged to obtain an overall measure for each time point with higher values indicating higher levels of PSU ($M_{T1}$ = 1.86, $SD_{T1}$ = .85, $M_{T2}$ = 1.94, $SD_{T2}$ = .86). In line with the work by Kwon and colleagues [64], the SAS-SV showed good levels of reliability at both waves ($\alpha_{T1}$ = .89 and $\alpha_{T2}$ = .89). The original items and the Italian translation of the scale can be found in the S3 Table.

**Social desirability.** Given that PSU was assessed through self-report, we accounted for adolescents' tendency to provide socially desirable answers in our multivariate analyses. Social desirability was measured at T1 with eight items from the Italian version of the Children's Social Desirability Short Scale [60]. Items were assessed on a 5-point scale ranging from 1 "never" to 5 "always". All items were reverse coded and averaged to obtain an overall measure of social desirability with higher values indicating higher social desirability levels (M = 2.64, SD = .70, α = .84).

## Analytical plan

After computing descriptive statistics for each smartphone use indicator, we, first, calculated the correlations between self-report data and trace data by considering both duration and frequency of smartphone use for a general day, weekdays, and weekend days. Since the data were not normally distributed, variables were log-transformed before they were entered in the

analyses. A Bonferroni correction was applied when self-report smartphone use was compared to traced duration and frequency of use (p ≤ .017).

Second, to obtain a measure of time distortion, we calculated a difference index Δ by subtracting self-report duration of use from automatically recorded use. Hence, negative values represent an over-estimation and positive values an under-estimation of smartphone use.

Third, we calculated Pearson's correlation coefficient for all the variables included in the subsequent regression models (i.e., for a general day, a weekday, and a weekend day). Additionally, we computed Spearman's rank correlations (by keeping each item in the original ordinal scale format) to better investigate how traced duration, frequency, and time distortion, were related to each item of the SAS-SV scale. The results were corrected for multiple comparisons (p ≤ .005).

Eventually, we ran regression models with PSU at T1 and T2 as the outcome variables predicted by traced duration and frequency of smartphone use as well as time distortion. Gender and social desirability were included as control variables. In the longitudinal model, we also included PSU at T1 to control for autoregressive effects. In particular, six regression models were run to predict PSU levels cross-sectionally (at T1) and one year later (at T2). Due to the small sample size, we decided to highlight marginally significant effects (p < .10) when reporting the regression results [65]. The dataset and the survey instrument are available in a repository at the following link: https://osf.io/hwr2u.

## Results

### Descriptive results

Duration of smartphone use was automatically collected through the *Ethica* application on participants' smartphones for 45 consecutive days from the enrollment date. Results of digital trace data showed that adolescents used the smartphone, on average, for one hour and fifty minutes per day, which was the average of all 45 recorded days irrespective of weekdays and weekend days (M = 1.89, SD = 1.52). The minimum was five minutes, and the maximum was six hours and fifty minutes. When looking at weekdays and weekend days separately, the recorded amount of time on a typical weekday was one hour and forty minutes (M = 1.68, SD = 1.51). In contrast, the average amount of time spent on the smartphone on a typical weekend day increased to two hours (M = 2.01, SD = 1.65). Trace data for frequency of use showed that adolescents activated the screen of their smartphones, on average, 57 times (median = 51) during a typical day, ranging from 7 to 222 times. The frequency of use tended to be lower on a weekday, with 55 times (median = 48, ranging from 7 to 196), on average, and higher on a weekend day (M = 66, median = 51, ranging from 1 to 286).

Regarding self-report measures, we found that participants indicated to spend approximately one hour and 53 minutes (M = 1.88, SD = 1.41) on their smartphone during a general day. However, they estimated to spend approximately one hour and forty minutes during a weekday (M = 1.67, SD = 1.40) and two hours and a half during a weekend day (M = 2.43, SD = 1.56). Also, participants reported somewhat higher levels of PSU at T2 compared to T1 ($M_{T1}$ = 1.86, $SD_{T1}$ = .85, $M_{T2}$ = 1.94, $SD_{T2}$ = .86). Yet, the increment was not statistically significant (t = -1.49, p = .140).

Additionally, based on the computed difference index (Δ), we estimated the extent of participants' time distortion when reporting on their smartphone use. The results showed that, in general, adolescents tended to equally over- and under-estimate, with the mean of the difference index being close to zero (M = .006, SD = 1.83). In particular, 44% of the sample tended to over-estimate their smartphone use on a general day, 43% tended to under-estimate, and 13% provided a correct smartphone use estimate (with an error ranging from -30 minutes to

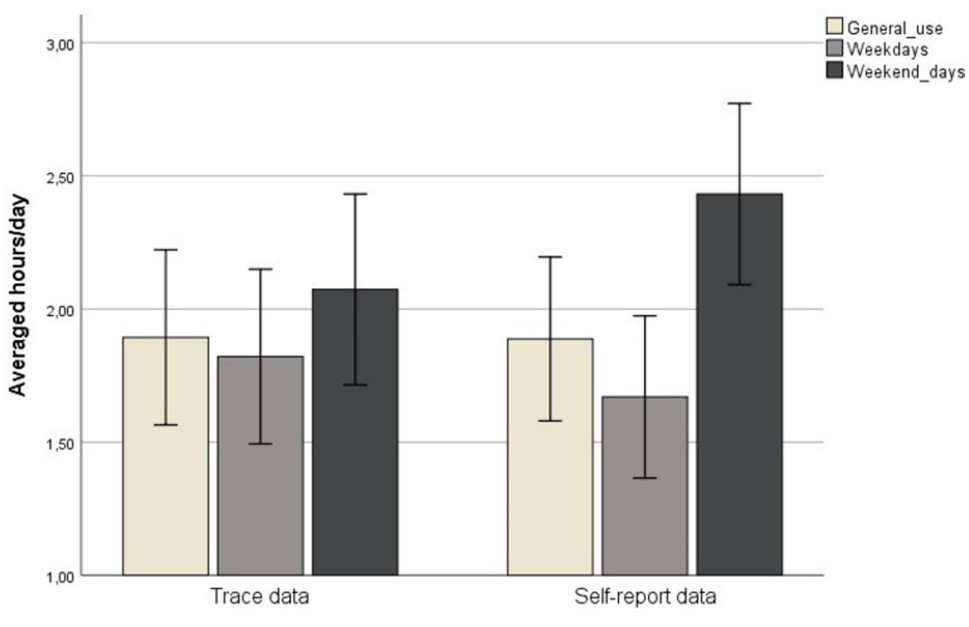

**Fig 1. Representation of traced and self-reported estimates (in hours per day) of the duration of smartphone use for a general day, a weekday, and a weekend day.**

+30 minutes). On weekdays, participants tended to under-estimate the time spent on the smartphone (M = .152, SD = 1.83; 38% over-estimators, 45% under-estimators, and 17% correct estimators). Conversely, during weekend days, adolescents tended to over-estimate smartphone use (M = -.358, SD = 2.02, with 54% over-estimators, 46% under-estimators, and 0% correct estimators). To note, the variability in time distortion was larger on a weekend day compared to a weekday. An overview of the results is shown in Fig 1.

## Accuracy of self-report measures of smartphone use

For a general day, self-report duration of smartphone use positively and significantly correlated with traced frequency of use (r = .276, p = .012). Similarly, for a typical weekday, self-report duration was correlated with traced frequency of use (r = .244, p = .027). After applying a correction for multiple comparisons, the result was no longer significant. However, looking at a typical weekend day, the self-report measure significantly correlated with traced frequency of use (r = .325, p = .014). No significant relationship was found between self-reports and traced duration of smartphone use for a general day, a typical weekday, and a weekend day. Thus, to answer RQ1, there was a small-to-medium relationship between self-report smartphone use and trace data, but only when the traced frequency of smartphone use was considered. With regards to RQ2, no noteworthy differences in the correlation coefficients were evident for self-report and traced smartphone use, neither on weekdays nor on weekend days.

## Preliminary correlations with problematic smartphone use

For a general day, Pearson's bivariate correlations showed that PSU at T1 correlated only with social desirability (r = .478, p < .001), whereas PSU at T2 correlated with time distortion, with higher PSU being associated with over-estimation (r = -.236, p = .035), and social desirability (r = .326, p < .001). For weekdays, PSU correlated only with social desirability at both T1 (r =

.478, p < .001) and T2 (r = .326, p < .001). Whereas, for weekend days, time distortion significantly and negatively correlated with PSU at both T1 (r = -.243, p < .001) and T2 (r = -.278, p < .001), in addition to social desirability (r = .478, p < -001 at T1 and r = .326, p < .001 at T2). Table 1 summarizes the results for a general day (see S1 and S2 Tables in the Supplement for weekdays and weekend days results).

Since the SAS-SV covers different facets of the concept of PSU (e.g., excessive use, withdrawal symptoms, conflict), we correlated each item of the SAS-SV scale with traced smartphone use and the difference index as an indicator of time distortion. We found that item 9 "Using my smartphone longer than I had intended" was the only item that was positively and significantly correlated with traced frequency of use (rho = .322, p = .003). Frequency of smartphone use also correlated with item 3 "Feeling pain in the wrists or at the back of the neck while using a smartphone" (rho = .297, p = .007) and item 5 "Feeling impatient and fretful when I am not holding my smartphone" (rho = .260, p = .017). However, after applying Bonferroni's correction, these latter results were no longer significant. Result are reported in S3 Table.

Furthermore, correlations between measures of smartphone use and PSU at T2 showed that time distortion negatively correlated with item 5 (rho = -.357, p = .001), meaning that over-estimators reported higher levels for this item. Time distortion also correlated with item 8 "Constantly checking my smartphone so as not to miss conversations between other people on Twitter or Facebook" (rho = -.221, p = .050). Traced duration of use was significantly related to item 10 "The people around me tell me that I use my smartphone too much" (rho = .226, p = .045), however the latter two were no longer significant after applying the correction for multiple comparisons. Results are reported in S4 Table.

## Regression results

Including gender, social desirability and measures of smartphone use, i.e. traced duration and frequency of smartphone use, and time distortion as predictors of PSU (see Table 2), regression results showed that social desirability (β = -.455, p < .001) and traced duration of smartphone use (β = .307, p = .038) were significantly associated with PSU at T1. Time distortion, i.e., the tendency to over-estimate smartphone use, showed only a marginally significant association with PSU at T1 (β = -.232, p = .062). In the longitudinal model, the tendency to over-estimate predicted significantly higher PSU levels one year later (β = -.267, p = .029), even after controlling for baseline levels of PSU.

**Table 1. Bivariate correlations among predictor and outcome variables for a general day.**

|  | 1. | 2. | 3. | 4. | 5. | 6. | 7. |
|---|---|---|---|---|---|---|---|
| 1. PSU at T1 | 1 |  |  |  |  |  |  |
| 2. PSU at T2 | .575** | 1 |  |  |  |  |  |
| 3. Trace duration of smartphone use | .150 | .102 | 1 |  |  |  |  |
| 4. Trace frequency of smartphone use | .199 | .179 | .580** | 1 |  |  |  |
| 5. Time distortion (Δ index) | -.176 | -.236* | .520** | .079 | 1 |  |  |
| 6. Gender | .001 | -.095 | -.062 | .137 | -.192 | 1 |  |
| 7. Social desirability | .478** | .326** | .071 | .323** | -.190 | -.122 | 1 |

Δ index represents traced duration minus self-report duration

*p < .05

** p < .001.

**Table 2. Regression results for predicting PSU at T1 and T2 from traced smartphone use on a general day.**

| Predictor variables | Problematic smartphone use at T1 | | Problematic smartphone use at T2 | |
|---|---|---|---|---|
| | B (S.E.) | β | B (S.E.) | β |
| 1.Gender | .038 (.085) | .046 | -.097 (.081) | -.116 |
| 2.Social desirability | **-.269 (.064)** | **-.455**\*\* | -.021 (.067) | -.035 |
| 3.Traced duration of smartphone use | **.116 (.055)** | **.307**\* | .050 (.055) | .132 |
| 4.Traced frequency of smartphone use | -.068 (.079) | -.114 | .036 (.076) | .060 |
| 5. Δ index | **-.053 (.028)** | **-.232**† | **-.062 (.028)** | **-.267**\* |
| 6.PSU at T1 | | | **.520 (.110)** | **.494**\*\* |
| Intercept | .108 (.28) | | .170 (.277) | |
| Adjusted-R$^2$ | .235 | | .344 | |
| F | 6.112 | | 7.89 | |
| p-value | < .001 | | < .001 | |

Δ index represents trace duration minus self-report duration

†p < .1

\*p < .05

\*\*p < .01.

When we considered the same predictor variables in the context of weekday estimates (see S5 Table), social desirability was significantly associated with PSU at T1 (β = -.262, p < .001). Traced duration of smartphone use was also related to PSU at T1 (β = .269, p = .060), but the association was marginally significant. In addition, PSU at T1 predicted the outcome at T2 (β = .508, p < .001). Time distortion also had a partially significant effect (β = -.233, p = .054).

Eventually, when changing the context to weekend days (see S6 Table), PSU at T1 was significantly associated with social desirability at T1 (β = -.436, p < .001), traced duration of smartphone use (β = .337, p = .044), and time distortion, in particular the tendency to overestimate (β = -.375, p = .005). Also, time distortion (β = -.302, p = .025) was a significant predictor of PSU at T2, controlling for PSU at T1 (β = .447, p < .001).

To conclude, H1 and H3 were generally sustained. Participants with a longer traced duration and an over-estimation of smartphone use also reported higher levels of PSU. Additionally, H2 was partially sustained, but only at the item level of PSU, i.e. for traced frequency of smartphone use and selected items of the SAS-SV.

## Discussion

The pervasiveness of digital media devices, especially smartphones, has raised concerns about adolescents' problematic use of these devices. Previous studies highlighted that PSU negatively impacts well-being [5,19,66,67]. However, the studies relied on self-report data to assess adolescents' (problematic) smartphone use, which is subject to several biases like recall, estimation, and social desirability bias [34]. Objectively recorded smartphone use overcomes limitations of self-reports [68], though the use of trace data poses challenges, too [47]. To date, few studies with adult samples combined self-report and digital trace data to study PSU [55]. However, data collected from underage populations are still scarce [14–16,53] and focused mainly on the frequency of calls and text messages, leaving out overall smartphone use like duration and frequency and other user behaviors such as social media use, streaming, gaming, or Internet use in general. Furthermore, past studies did not relate trace data to PSU.

To the best of our knowledge, this is the first study using digital trace data to explore predictors of PSU in adolescence. Using a dedicated application installed on participants' devices, we collected trace data from a sample of 84 adolescents aged 13 to 14 years and compared objective trace data to self-reported estimates. We, furthermore, used both trace data and self-report data to identify predictors of higher PSU levels, both cross-sectionally and longitudinally. Our results provided valuable insights into smartphone users' dynamics and disentangled predictors of PSU in adolescence. While some results were in line with past research on adult populations, others were not.

In contrast to previous literature on the adult population, which showed inconsistent results on the accordance between self-report and trace data [37,66–68], we found that trace and self-report data showed similar values for smartphone use. Thus, adolescents tended to report a relatively reliable estimate of their time spent with the smartphone *per* day, which, in our sample, amounted to approximately two hours. However, looking at the correlations between trace and self-report data, coefficients were significant only for self-reported *duration* of smartphone use and traced *frequency* of smartphone use, not traced duration. This result was consistent for estimates referring to a general day, a weekday, and a weekend day. In other words, when adolescents were asked to estimate the total time (duration) they spent with their smartphones on a typical weekday and weekend day, they relied more on the frequency of use, i.e. the frequency of their checking behaviors. This result highlights an important point to consider when adolescents report on their online behaviors: The cognitive process involved in estimating the duration of smartphone use is determined by the interference (approximated by the frequency of checking behaviors) of the smartphone in adolescents' everyday activities, and less by the duration of the use. In fact, checking the smartphone, especially in repeated short time intervals, is intrusive and distracting and, thus, more salient when asked to report on the duration of smartphone use.

The interference of the smartphone with everyday activities is also an indicator of PSU. As such, we found that the traced frequency of smartphone use in the present study correlated with item 9 of the SAS-SV scale, which tackles a subdimension of PSU related to the perception of excessive time spent on the smartphone. Without correcting for multiple comparisons, the frequency of smartphone use also correlated with the other two items of the same scale, thus further underlying the link between the two variables. Conversely, traced duration of smartphone use did not correlate with any item of the PSU scale.

The predictive role of the frequency of smartphone use is not surprising, considering the nature of checking behaviors. Indeed, checking habits are specific to smartphone use, rather than the use of other devices like, for example, laptops and tablets [69]. Smartphones offer quick and easy access to rewards and work as a gateway to other applications and activities, thus increasing the overall time spent on the device and interfering with daily life. As reported by Heitmayer and Lahlou [18], young people described picking up their phones as "feeling natural or automatic, and even unconscious *like when you cough and put your hand over your mouth*" (p. 5). Interestingly, the authors also found that almost all study participants picked up, unlocked, used, and put back the phone without a particular purpose in mind—an activity named "fidgeting". Additionally, when unlocked, young people often fall into a *loop*. For example, the engagement in one social media app also triggers the use of other social media platforms, thus making participants spend more time on the smartphone than initially intended. Considering our results, it is possible that adolescents with higher PSU levels also unlock and use the phone without a particular intention multiple times a day, thus entering into a loop.

However, when predicting PSU, the traced duration of smartphone use, rather than the traced frequency of use, showed a significant relationship. In other words, *traced frequency* is

associated with *self-reported duration* of smartphone use, whereas *traced duration* is related to *self-reported PSU*.

This result is partially in line with findings from tracking studies in adult populations. For example, Montag and colleagues [70] found that both phone use recorded in hours per week and the number of calls and text messages were related to problematic mobile phone use. Also, Noë and colleagues [59] showed that overall usage and scrolling events were positively associated with PSU (with social media apps like Instagram being a primary source of activity for smartphone addicts). Tossell and colleagues [58] underlined how smartphone addict users spent twice as much time on the phone and started interaction with applications twice as often than non-addict users. Our results also align with a recent meta-analysis showing a small correlation between PSU and traced smartphone use [71]. On the contrary, Lin and colleagues [50] found that the frequency of checking behaviors, and not the duration of smartphone use, was related to PSU. Our findings are also consistent with findings from research relying on self-reports only. Several studies showed that a longer duration of smartphone use was associated with higher levels of PSU [72–74].

In general, smartphone use may obstruct everyday routines and specific activities such as studying/learning, working, and offline social interactions [75]. Scales measuring PSU pick up this aspect and include items such as "Having a hard time concentrating in class, while doing assignments, or while working due to smartphone use" or "Constantly checking my smartphone so as not to miss conversations between other people on Twitter or Facebook" [23]. The interference due to smartphone use creates functional impairments, which are similar to the ones caused by other behavioral addictions, but, at the same time, also specific to PSU [50]. In accordance with this hypothesis, using the smartphone in an *absent-minded* way has been found to have a pervasive and strong positive link with various measures of inattention [76]. Furthermore, prior research has identified impulsivity and the urgency to immediately check incoming notifications as two risk factors for PSU [77,78]. In addition, fear of missing out, i.e., the "pervasive apprehension that others might be having rewarding experiences from which one is absent" [79, p. 1841] has been related to PSU [80–82].

A second major finding is that adolescents tend to both over- and under-estimate their smartphone use. This result is in line with a recent meta-analysis [71] showing that only three out of the 49 included comparisons showed that self-reported media use was close to the logged mean, whereas an equal number of studies showed that participants either over- (k = 23) or under- (k = 23) estimated their time spent using digital media devices. To note, in our study, time distortion was related to PSU both cross-sectionally and longitudinally. Additionally, over-estimation was related to item 5 of the SAS-SV scale, which tackles the experience of withdrawal symptoms (e.g., irritability) when one cannot use the smartphone. More precisely, our study revealed that over-estimation predicted higher PSU levels over time, even after controlling for social desirability and baseline levels of PSU. One can conclude from this finding that adolescents who excessively used their smartphones were well aware of their problematic use and reported a higher duration of smartphone use. However, based on what we just said, adolescents seemed to rely more on their perceptions of the *frequency* of use, i.e. how much they repeatedly check their smartphones, rather than of the *duration* of use when reporting on the latter. This explains, in part, the role of over-estimation of smartphone use in relation to PSU, as shown in the significant effect of the difference index in the model for a general day and a typical weekend day.

Although our findings of over-estimation contradict results from a prior study by Lin and colleagues [50], who found that under-estimators had higher PSU levels, they are in line with previous research on social media and online gaming. For example, Turel and colleagues [83] found an effect of time distortion on social media addiction. In particular, the at-risk group

showed a significant over-estimation bias, which positively correlated with the Facebook addiction scale; in contrast, the no-risk group tended to report an under-estimation of social media use. Furthermore, our results aligned with a recent study investigating social media use in adolescents [84], which also found that participants tended to over-estimate the time spent on social media platforms, with a more accurate estimation for Instagram use rather than the use of WhatsApp or Snapchat.

Consistent with different perception theories [85,86], time distortion can be a cognitive marker that may help classify youth in 'at-risk' and 'no-risk' groups concerning media-related addictions. From a cognitive perspective, over-estimation can be related to impaired attention mechanisms since people are engaging in highly rewarding behaviors, e.g., social networking or online gaming [83,87]. The involvement in this kind of activities may over-emphasize the perception of time spent doing them. Furthermore, over-estimation can be related to a cognitive distortion of short durations (which are often over-estimated) and long durations (which are often under-estimated), following Vierordt's law [88–90]. Thus, frequent checking behaviors that last for short moments may be estimated, overall, as a longer (daily) use of the device than what it actually is.

A third important finding is that, in contrast to past studies that considered general smart-phone use, our results added valuable insights on the differences and similarities in self-reported and traced smartphone use on *weekdays* and *weekend days*, and their role in PSU. However, future research should further investigate smartphone use in different days of the week characterized by more or less structured daily routines (e.g., school on weekdays, free time on weekend days). That would shed light on how such routines play a role in estimating smartphone use and predicting a problematic use of the device.

Eventually, it should be noted that social desirability bias plays an important role in predict-ing higher levels of PSU in adolescents. More precisely, we found that participants who were more inclined to provide socially desirable answers reported lower PSU levels. This finding may not be surprising, considering that social desirability describes respondents' tendency to inaccurately report on sensitive topics, in order to present themselves in the best possible light [39]. When sensitive topics have a negative connotation, as in the case of PSU, adolescents with a higher tendency to provide socially desirable answers adapted their responses to what they think the other person (i.e. researcher) considers as socially acceptable. Thus, they tended to under-report when asked about (problematic) smartphone use, for example, how often they have a hard time concentrating in class or how often they miss planned work due to their smartphone use, or how often they use the smartphone longer than intended. While digital trace data allow accounting for a potential estimation bias, as demonstrated in the present study, the reliance on automatically recorded data is not sufficient to account for a possible social desirability bias inherent to self-reported measures such as those typically applied to assess PSU. Critics have repeatedly argued that participants with higher levels of social desir-ability propensity may adapt their habitual use of the smartphone when being aware that they are traced. The present study objectively collected data on adolescents' smartphone use for 45 consecutive days to avoid biases due to short term adaptations of habitual smartphone use. Yet, other studies with adult populations used much shorter timeframes, arguing that a few days are enough to predict repetitive behaviors reliably [83].

## Limitations and recommendations for future research

Our study comes with some limitations that should be acknowledged. In particular, due to the small sample size, our statistical analyses should be interpreted with caution, and future studies should replicate our findings with larger samples. Additionally, samples should represent the underlying population, which is a noteworthy challenge because many adolescents see digital

trace data as a strong invasion of their privacy [91]. Privacy concerns are also a challenge for participants' adherence over time, which is lower than in traditional survey-based studies, and only a few people are generally predisposed to donate log data [92].

Furthermore, we faced some technical problems due to the *Ethica* application malfunctions or incompatibility with older smartphone models. Although the application provider partially resolved these problems, they still resulted in dropouts, reducing even further our analytical sample. Besides, trace data collected as part of this study was limited to the overall duration and frequency of smartphone use. Thus, we have no information on what activities adolescents engaged in on their smartphones based on objectively recorded data. Although ethical and legal restrictions limit the possibility to obtain objective data on specific contents, application usage can be integrated into tracking studies. Future studies should thus focus on specific application usage, with a particular emphasis on social networking applications [93] and online gaming, given that both are popular activities among younger populations [94]. Finally, by investigating frequency, duration, and type of applications used, future research should rely more on objective criteria or expert ratings of PSU, as partially done in research on adult populations [50], to overcome the social desirability bias problem, thus providing a starting point for (smartphone-based) interventions to reduce PSU. This may further allow enhancing our knowledge of drivers and indicators of PSU.

Additionally, PSU was the only self-report measure of well-being related to trace smartphone use in this study. However, several other factors could impact the study's findings, for example, the presence of comorbid mental disorders like subthreshold symptoms of depression and anxiety or personality traits like neuroticism and impulsivity [95]. Since the present study did not consider these additional factors in explaining the link between trace and self-report smartphone use, future research should further identify vulnerable adolescent populations characterized by mental disorders or dysfunctional traits to investigate differential relationships between predictors of PSU based on digital trace and self-report data.

Finally, on a side note, although the SAS-SV used in the present study has been found to predict PSU levels based on experts' diagnoses [23], its validation has not been carried out in clinical settings. Hence, the SAS-SV can be only used to detect potentially high-risk groups of PSU in the general population. To overcome these limitations, future studies should complement self-assessments of PSU with clinical interviews to further validate the use of this scale in assessing symptoms of PSU.

## Conclusions

PSU remains a critical concept to be studied in adolescents due to the identified negative short- and long-term consequences for personal well-being and development. Valid and reliable predictors and PSU indicators are urgently needed, and digital trace data proved to be an informative data source that requires further investigations to eventually develop diagnostic tools and indices of PSU that incorporate trace data.

## Supporting information

**S1 Table. Bivariate Pearson's correlations among predictor and outcome variables for a weekday.**
(DOCX)

**S2 Table. Bivariate Pearson's correlations among predictor and outcome variables for a weekend day.**
(DOCX)

**S3 Table. Spearman's rank correlations between items of the SAS-SV at T1, traced duration and frequency of smartphone use, and difference index.**
(DOCX)

**S4 Table. Spearman's rank correlations between items of the SAS-SV at T2, trace duration and frequency of smartphone use, and difference index.**
(DOCX)

**S5 Table. Regression results for PSU at T1 and T2 including smartphone use on a weekday.**
(DOCX)

**S6 Table. Regression results for PSU at T1 and T2 including smartphone use on a weekend day.**
(DOCX)

## Acknowledgments

We would like to thank the collaborating schools for their help in student recruitment and data collection. We also thank Mohammad Hashemian and Chiara Antonietti for their help during the preparation and data collection phase as well as Nathaniel Osgood and Teena Thomas during the data analysis phase.

## Author Contributions

**Conceptualization:** Laura Marciano, Anne-Linda Camerini.

**Data curation:** Laura Marciano.

**Formal analysis:** Laura Marciano.

**Funding acquisition:** Anne-Linda Camerini.

**Methodology:** Laura Marciano.

**Project administration:** Anne-Linda Camerini.

**Supervision:** Laura Marciano, Anne-Linda Camerini.

**Writing – original draft:** Laura Marciano.

**Writing – review & editing:** Laura Marciano, Anne-Linda Camerini.

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
