## [Decision Letter · Decision Letter 0]

5 Jul 2021

PONE-D-21-09423

Duration, frequency, and time distortion: Which is the best predictor of problematic smartphone use in adolescents? A trace data study

PLOS ONE

Dear Dr. Marciano,

Thank you for submitting your manuscript to PLOS ONE. After careful consideration, we feel that it has merit but does not fully meet PLOS ONE’s publication criteria as it currently stands. Therefore, we invite you to submit a revised version of the manuscript that addresses the points raised during the review process.

In particular, one reviewer raised serious concerns regarding the methology and statistical analyses of your manuscript. Please pay specific attention to these comments since methodological concerns should be targeted and alleviated during a revision.

We look forward to receiving your revised manuscript.

Kind regards,

Michael Kaess, M. D.

Academic Editor

PLOS ONE

Journal Requirements:

3. Thank you for including your ethics statement: "The Cantonal education administration of Ticino approved the annual panel study based on self-administered questionnaires. The embedded Ethica study received approval from the Ethics Committee of the university where the research was carried out and the Cantonal Data Protection Officer. Participation in the Ethica study required active consent by parents who had to send a signed consent form to the Cantonal education administration, which, in turn, provided the research team with the associated U-ID. Students provided their active consent directly in the Ethica application upon enrolment."

4. Please include additional information regarding the survey or questionnaire used in the study and ensure that you have provided sufficient details that others could replicate the analyses. For instance, if you developed a questionnaire as part of this study and it is not under a copyright more restrictive than CC-BY, please include a copy, in both the original language and English, as Supporting Information.

Reviewers' comments:

Reviewer's Responses to Questions

**Comments to the Author**

1. Is the manuscript technically sound, and do the data support the conclusions?

Reviewer #1: Partly

Reviewer #2: No

Reviewer #3: Yes

2. Has the statistical analysis been performed appropriately and rigorously? 

Reviewer #1: No

Reviewer #2: No

Reviewer #3: Yes

3. Have the authors made all data underlying the findings in their manuscript fully available?

Reviewer #1: No

Reviewer #2: Yes

Reviewer #3: Yes

4. Is the manuscript presented in an intelligible fashion and written in standard English?

Reviewer #1: Yes

Reviewer #2: Yes

Reviewer #3: Yes

5. Review Comments to the Author

Reviewer #1: A timely and relevant topic, innovative methods, interesting findings, and overall, a well-written manuscript that I enjoyed reading. Nevertheless, some issues deserve further attention. There are three major hypotheses tob e tested and two further research questions. And some further approaches, such as dividing the sample into „light vs heavy users“. I’d encourage the authors to specify the signficance threshold for their hypotheses and to state whether correction for multiple comparisons was performed or not. Also, further approaches, as referred to above, need to be specified in the methods and not in the results section. Several times in the manuscript the paper refer tot he exploratory nature of their analysis. While I appreciate this transparency, there are nevertheless hypotheses which were explicitly tested, and several other parameters that were subject to exploratory analyses. I’d encourage the authors to differentiate between both approaches, to state whether a result survived testing for multiple comparisons or not, and to be very cautious in their approach to „also highlight marginally significant effects“ (p. 13), I’m afraid that this could be at times misleading when truly robust effects should be identified from a plethora of analyses performed in this study. Also, the limitations section needs some further attention. Only relationships with the SAS scale were used, and no comparisons with other instruments were made. Several factors that could have an impact on this study’s findings were not considered, e.g. the impact of comorbid mental disorders (if any in this sample, yet I couldn’t find a statement regarding this issue), subthreshold depression or anxiety levels or personality traits, e.g. impulsivity. Finally, although I’m aware that this may further increase the number of analyses, regression models (at least for a specific limited number of variables) could be considered for the SAS subscales reported by Kwon and colleagues in 2013. This information may beneficially complement the paper as supplementary data.

Reviewer #2: PONE-D-21-09423_reviewer

Review on “Duration, frequency, and time distortion: Which is the best predictor of problematic smartphone use in adolescents? A trace data study”. The authors used subjective and objective measures of smartphone use at two assessments with a one year interval and examined the association between the subjective and objective measures, as well as the associations between these measures and the problematic smartphone use score. The study was conducted in a rather small sample of adolescents (n=84). In my opinion, the study has some major methodological issues, which make it impossible to interpret the data. Most importantly, the scale of the self-report measure and the conversion of the objective measures to match this scale, and the statistical analyses renders it hard to interpret the results. Moreover, in my opinion a Bland-Altman tests is the method to use when trying to compare two assessments (and not multiple correlation coefficients). In light of these major critics and because the problem with the scale is baked into the data, all most of my other comments listed below are minor issues.

Lines 49 – 51: However, until today, most of the studies focusing on PSU relied on self-report data, making it hard to draw valid conclusions on which type of smartphone activity can be described as problematic. – to make a stronger argument I would suggest to add references indicating that the validity of self-reports of use are questionable

Line 52: Could you please provide a definition of “digital trace data”? A short definition is sufficient.

Lines 63 – 68: Those are large percentages – however, the data stems from a diverse group. I would suggest that the very high numbers in the 90s are due to the large span of years of age of the participants (12 to 19 years). Could you provide more details on differences between the age groups? I.e., how the percentage is associated with years of age?

There are some redundancies because of duplicates in the paragraphs „Digital trace data“ and „Ethical considerations“; please stream line by deleting these redundancies

The „Sample“ paragraph clearly describes the reduction of the sample and the causes why only 84 out of the original panel of 1419 persons. Could you please provide some information whether there are differences between those who participated and those who did not? Please discuss

Measures: The scale of the self-rated use of the smartphone is not suitable to calculate a mean since it is not measured on an interval scale (the intervals between two ratings are not the same)!

Line 251: I think the results assessed by the scale should be presented in the results section and not in the method section. The same applies to the other two trace data measures. Moreover, what does is the correlation coefficient presented for each measure? The correlation between weekday use and weekend use? It is not addressed and not obvious to the readers.

Lines 274 and following: Again, I would refrain from presenting the results in the method section. Moreover, I would suggest to point out the psychometrics of the Smartphone Addiction Scale for adolescents (the alpha is presented within the brackets showing the means and SDs…).

Analytic plan: I think it is problematic to report marginally significant results and, on top of that, calculate multiple tests without correcting for the multiple testing.

Lines 319 and following: In my opinion, it is not possible to interpret the average scores of the subjective rating and you shouldn’t report or interpret the scores in the way you did. Moreover, which correlation coefficient did you calculate?

Table 1: You report a lot (a lot!) of correlation coefficients here. I would suggest to use a Bonferroni correction or another method accounting for the multiple tests that you conducted (and omit the marginal significant results!). Most important: If you converted the trace data into the same scale as the self-report measure, the data is not suitable for calculating means and SDs!

Lines 331 through 342: I think the statistical procedure is not adequate and therefore it is not possible to the interpret the results (and review this section).

Lines 344 and following: Duration of smartphone use and PSU – Did you use the trace data of the first assessment to divide the sample into light and heavy users? Please describe your procedure in more detail.

Besides the problems with the scale, I think categorizing the data can often be misleading. I follow your rational on the differentiation between those with more than 2 hr screen time and those with less – but for the other categorizations I would use the full information available and do not just compare those categories.

Regrading the regression analyses: have you checked for multicollinearity in your models? Can you please provide more information on that?

Table 3. Did you use the categorized variables in your regression analyses? I’m wondering because you list being part of the light users group/the under-estimators group/the low checkers group as references.

Minor issues:

Line 80: Please correct the references and delete the duplicate (10,20,20,21)

Please correct typos, e.g. Internet (line 156), line 455 (64,65,38,66),…

Reviewer #3: The paper is concisely written and addresses an important topic – the validity of indiciators of problematic smartphone use – assessed via digital trace data. Particularly the addressed sample of adolescents is very interesting for early interventions and thus, data derived from this population group are valuable. Unfortunately, only a very small subsample (89 out of 1419 students) could be recruited for the current study.

• To assess the representativeness of the sample and the results, it would be interesting to compare the reached subsample with the total sample of the Ethica study.

• The authors argue that self-report data on PSU might not be reliable but validate their digitally traced data also with data from a self-report measure on Smartphone Use (SAS-SV). What about the validity of this instrument? The limitations of this approach should be discussed. Why did you not conduct clinical interviews to assess PSU.

• The results on the indicator "frequency of checking" are of high interest, I wonder why the authors did not include a measure analoguous to "Perceived duration of smartphone use" to assess "perceived frequency of checking".

6. PLOS authors have the option to publish the peer review history of their article (what does this mean?). If published, this will include your full peer review and any attached files.

Reviewer #1: No

Reviewer #2: No

Reviewer #3: No

---

## [Author Response · Author response to Decision Letter 0]

16 Aug 2021

Comments to the Author

1. Is the manuscript technically sound, and do the data support the conclusions?

Reviewer #1: Partly

Reviewer #2: No

Reviewer #3: Yes

2. Has the statistical analysis been performed appropriately and rigorously?

Reviewer #1: No

Reviewer #2: No

Reviewer #3: Yes

3. Have the authors made all data underlying the findings in their manuscript fully available?

Reviewer #1: No

Reviewer #2: Yes

Reviewer #3: Yes

4. Is the manuscript presented in an intelligible fashion and written in standard English?

Reviewer #1: Yes

Reviewer #2: Yes

Reviewer #3: Yes

5. Review Comments to the Author

Reviewer #1: A timely and relevant topic, innovative methods, interesting findings, and overall, a well-written manuscript that I enjoyed reading. 

A: Many thanks for the comment. We appreciate that you enjoyed the manuscript. 

Nevertheless, some issues deserve further attention. There are three major hypotheses tob e tested and two further research questions. And some further approaches, such as dividing the sample into „light vs heavy users“. 

I’d encourage the authors to specify the signficance threshold for their hypotheses and to state whether correction for multiple comparisons was performed or not. 

A: Many thanks for the comment. In the current version, as also suggested by Reviewer 2, we removed the categorizations of high versus low users resp. checkers, and the categories of over- and under-estimation. Instead, we used the continuous scales. Additionally, we corrected the significance level of correlations by using Bonferroni’s post-doc test for multiple comparisons. 

Also, further approaches, as referred to above, need to be specified in the methods and not in the results section. 

A: Many thanks. We reformulated the Analytical plan subsection in the methods section and the Result section (which now includes the following sub-sections: “Descriptive results”, “Accuracy of self-report measures of smartphone use”, “Preliminary correlations with problematic smartphone use”, and “Regression results”). 

Several times in the manuscript the paper refer tot he exploratory nature of their analysis. While I appreciate this transparency, there are nevertheless hypotheses which were explicitly tested, and several other parameters that were subject to exploratory analyses. I’d encourage the authors to differentiate between both approaches, to state whether a result survived testing for multiple comparisons or not, and to be very cautious in their approach to „also highlight marginally significant effects“ (p. 13), I’m afraid that this could be at times misleading when truly robust effects should be identified from a plethora of analyses performed in this study. 

A: Many thanks for the comment. We now applied a Bonferroni correction for multiple comparisons when needed. We highlighted marginally significant p-values only for the regression results. We think that marginally significant regression results deserve mentioning, since our analyses are of explorative nature and no other study exists on the relationship between trace smartphone use and PSU in adolescence, thus we cannot ignore a significant trend which can be informative for future research. 

Also, the limitations section needs some further attention. Only relationships with the SAS scale were used, and no comparisons with other instruments were made. Several factors that could have an impact on this study’s findings were not considered, e.g. the impact of comorbid mental disorders (if any in this sample, yet I couldn’t find a statement regarding this issue), subthreshold depression or anxiety levels or personality traits, e.g. impulsivity. 

A: Many thanks for the comment, which we valued. However, we could not consider other instruments as they were not assessed as part of this study. Thus, we added the following sentence in the Limitations section: “Additionally, SP addiction was the only self-report measure of well-being related to trace SP use in this study. However, several other factors could impact the study’s findings, e.g. the presence of comorbid mental disorders like subthreshold symptoms of depression and anxiety or personality traits like neuroticism and impulsivity. Although the present paper did not consider these additional factors in explaining the link between traced and self-reported data, future research should further identify vulnerable adolescent populations…”

Finally, although I’m aware that this may further increase the number of analyses, regression models (at least for a specific limited number of variables) could be considered for the SAS subscales reported by Kwon and colleagues in 2013. This information may beneficially complement the paper as supplementary data.

A: Many thanks for the comment. We added Spearman’s correlation coefficient between trace data and each item of the SAS-SV scale at both T1 and T2, controlling for multiple comparisons, as reported in Table 3 and 4 of the Appendix. We also reported on the main significant results in the Results section of the manuscript (p. 17) and discussed them in the Discussion section (p. 22 and p.24). 

We hope that our revisions are satisfactory and are available for any further questions you may have.

Reviewer #2: PONE-D-21-09423_reviewer

Review on “Duration, frequency, and time distortion: Which is the best predictor of problematic smartphone use in adolescents? A trace data study”. The authors used subjective and objective measures of smartphone use at two assessments with a one year interval and examined the association between the subjective and objective measures, as well as the associations between these measures and the problematic smartphone use score. The study was conducted in a rather small sample of adolescents (n=84). In my opinion, the study has some major methodological issues, which make it impossible to interpret the data. 

Most importantly, the scale of the self-report measure and the conversion of the objective measures to match this scale, and the statistical analyses renders it hard to interpret the results. Moreover, in my opinion a Bland-Altman tests is the method to use when trying to compare two assessments (and not multiple correlation coefficients). In light of these major critics and because the problem with the scale is baked into the data, all most of my other comments listed below are minor issues.

A: Many thanks for the comment. We acknowledge that using Bland-Altman tests would be beneficial when comparing two measures of the same constructs. In our case, that would be possible only for the duration of smartphone use but not for the frequency of use. Hence, we thank the reviewer for the suggestion, but we decided to rely on correlation analyses as done in comparable studies (see for example Verbeij, T., Pouwels, J. L., Beyens, I., & Valkenburg, P. M. (2021). The accuracy and validity of self-reported social media use measures among adolescents. Computers in Human Behavior Reports, 3, 100090. https://doi.org/10.1016/j.chbr.2021.100090), but changed the conversion of measures. More specifically, we now converted the self-reported measure of smartphone use by using the midpoint for each interval as a proxy of smartphone use, (e.g., 15 minutes for “0 to 30 minutes”. For the highest interval (“5 or more hours”), we used 5.5 hours as a proxy of time spent with the smartphone. This approach allowed us to consider the continuous scale of traced smartphone use and study its relation to self-report use and PSU. 

Lines 49 – 51: However, until today, most of the studies focusing on PSU relied on self-report data, making it hard to draw valid conclusions on which type of smartphone activity can be described as problematic. – to make a stronger argument I would suggest to add references indicating that the validity of self-reports of use are questionable

A: Many thanks for the suggestion. We added the following citation to the sentence: Verbeij T, Pouwels JL, Beyens I, Valkenburg PM. The accuracy and validity of self-reported social media use measures among adolescents. Comput Hum Behav Rep [Internet]. 2021 Jan 1 [cited 2021 Jun 30];3:100090. Available from: https://www.sciencedirect.com/science/article/pii/S2451958821000385

Line 52: Could you please provide a definition of “digital trace data”? A short definition is sufficient.

A: Thanks for the comment, we added the requested definition (lines 54-55). 

Lines 63 – 68: Those are large percentages – however, the data stems from a diverse group. I would suggest that the very high numbers in the 90s are due to the large span of years of age of the participants (12 to 19 years). Could you provide more details on differences between the age groups? I.e., how the percentage is associated with years of age?

A: Many thanks for the comment. We added more information with respect to the type of smartphone use for different age ranges as available in the original report (lines 70-75). 

There are some redundancies because of duplicates in the paragraphs „Digital trace data“ and „Ethical considerations“; please stream line by deleting these redundancies

A: Many thanks for the comment. We removed the redundant parts. 

The „Sample“ paragraph clearly describes the reduction of the sample and the causes why only 84 out of the original panel of 1419 persons. Could you please provide some information whether there are differences between those who participated and those who did not? Please discuss

A: Many thanks for the comment. We added the following sentence (lines 239-242): 

“With respect to students who did not join the Ethica study, participants did not report any difference in gender (p=.205), perceived socioeconomic status (p=.229), or self-reported daily smartphone use (p=.114).” 

More information can be found here: (Camerini, A.-L., & Marciano, L. (2019, July 15). Self-selection bias in research including ecological momentary assessment and digital trace data. Passive Smartphone Data Collection and Additional Tasks in Mobile Web Surveys: Willingness, Non-Participation, Consent, and Ethics I. 8th ESRA Conference, Zagreb.)

Measures: The scale of the self-rated use of the smartphone is not suitable to calculate a mean since it is not measured on an interval scale (the intervals between two ratings are not the same)!

A: Thanks for the observation. We decided to convert the original interval scale into hours. We outlined this decision in the text under the description of Perceived duration of smartphone use (lines 255-262): 

“To allow the comparison with trace data, results were converted into hours by using the midpoint for each category of the original interval scale: (0=0) (1=0.25) (2=0.75) (3=1.25) (4=1.75) (5=2.5) (6=3.5) (7=4.5) (8=5.5). For the highest interval (“5 or more hours”), we used 5.5 hours as a proxy of time spent with the smartphone.” (see Mößle, T. (2012). Dick, dumm, abhängig, gewalttätig? Nomos Verlagsgesellschaft mbH & Co. KG.)

By doing so, the comparison with the trace data of duration is now more straightforward and precise since both are expressed in the numeric equivalent of hours. 

Line 251: I think the results assessed by the scale should be presented in the results section and not in the method section. The same applies to the other two trace data measures. 

A: Many thanks for the comment. We reformulated the Analytical plan in the Methods section and the Result section (which now includes the following sub-sections: “Descriptive results”, “Accuracy of self-report measures of smartphone use”, “Preliminary correlations with problematic smartphone use”, and “Regression results”).

Moreover, what does is the correlation coefficient presented for each measure? The correlation between weekday use and weekend use? It is not addressed and not obvious to the readers.

A: We agree that reporting many correlations coefficients can be misleading. Hence, we removed the correlation table and reported only the Pearson’s correlations between smartphone use on a general day, weekdays, and weekend days, without mixing the results of weekdays and weekend days. We also applied a Bonferroni correction for multiple comparisons. All the variables were log-transformed before the analyses. 

We reported the results in the text under the new section “Accuracy of self-report measures”. 

Lines 274 and following: Again, I would refrain from presenting the results in the method section. Moreover, I would suggest to point out the psychometrics of the Smartphone Addiction Scale for adolescents (the alpha is presented within the brackets showing the means and SDs…).

A: Many thanks for the suggestion. We pointed out the values of Cronbach’s alpha in the text (lines 286-288). We also added a paired-sample t-test to assess if PSU changed one year later (line 344). 

Analytic plan: I think it is problematic to report marginally significant results and, on top of that, calculate multiple tests without correcting for the multiple testing.

A: We now applied Bonferroni correction when correlating trace and self-report data (both self-report smartphone use and PSU). Additionally, regarding the choice to comment on marginally significant results of the regression analyses, we would like to emphasize that trends in social sciences like psychology reported an increase in the commentary of marginal effects, since sometimes “A marginal result might be interpreted as a caution against “accepting” the null hypothesis, a promising preliminary result, or sufficient evidence for some noncentral hypothesis, or it might even be interpreted as equivalent to a significant result” and the .05 cut-off is essentially arbitrary (Pritschet, L., Powell, D., & Horne, Z. (2016). Marginally Significant Effects as Evidence for Hypotheses: Changing Attitudes Over Four Decades. Psychological Science, 27(7), 1036–1042. https://doi.org/10.1177/0956797616645672). Since there are no official guidelines on how to report marginally significant results, but only many different opinions, we would prefer to highlight and not dismiss them, since they might be useful for future research on the same topic. 

Lines 319 and following: In my opinion, it is not possible to interpret the average scores of the subjective rating and you shouldn’t report or interpret the scores in the way you did. Moreover, which correlation coefficient did you calculate?

A: Many thanks for the comment. As stated before, to allow the comparison with trace data, self-report data were converted into hours by using the midpoint for each interval as a proxy of smartphone use, (e.g., 15 minutes for “0 to 30 minutes”. For the highest interval (“5 or more hours”), we used 5.5 hours as a proxy of time spent with the smartphone. This approach allowed us to consider smartphone use as a continuous scale and calculate Pearson’s correlation coefficients for self-report and trace data. 

Table 1: You report a lot (a lot!) of correlation coefficients here. I would suggest to use a Bonferroni correction or another method accounting for the multiple tests that you conducted (and omit the marginal significant results!). 

A: Many thanks for the comment. We agree that reporting many correlation coefficients can be misleading. Hence, we removed the correlation table and reported only the Pearson’s correlations between smartphone use on a general day, weekday and weekend day, without mixing the results of weekdays and weekend days.

Most important: If you converted the trace data into the same scale as the self-report measure, the data is not suitable for calculating means and SDs!

A: Many thanks for the comment. As stated above, to allow the comparison with trace data, self-report data for smartphone use were now converted into hours using the midpoint of each interval.

Lines 331 through 342: I think the statistical procedure is not adequate and, herefore, it is not possible to interpret the results (and review this section).

A: Thanks to the new approach to convert self-report smartphone use into hours, we now changed this part and reported on the correlation coefficients, corrected for multiple comparisons (see lines 360 to 371).

Lines 344 and following: Duration of smartphone use and PSU – Did you use the trace data of the first assessment to divide the sample into light and heavy users? Please describe your procedure in more detail.

A: Many thanks for the comment. Trace data were collected only at T1. However, we now decided to avoid categorizing variables and we used the continuous variables for our analyses. 

Besides the problems with the scale, I think categorizing the data can often be misleading. I follow your rational on the differentiation between those with more than 2 hr screen time and those with less – but for the other categorizations I would use the full information available and do not just compare those categories.

A: Many thanks for the comment, which we valued. Hence, we decided to use the full information available for all continuous variables and to report the results (i.e., correlation and regression results) for a general, a weekday, and a weekend day. The tables for a general day are placed in the main manuscript, whereas the tables for a weekday and a weekend day are reported in the supplement file. All the variables were log-transformed before the analyses to avoid problems of linearity. 

Regrading the regression analyses: have you checked for multicollinearity in your models? Can you please provide more information on that?

A: many thanks for the comment. We added the correlation table for each regression analysis (in which it is possible to see that the correlations among trace data were still lower than r=.80 as a cut-off proposed by Berry WD, Feldman S. Multiple Regression in Practice (Quantitative Applications in the Social Sciences) SAGE Publications; Thousand Oaks. CA: 1985). 

Additionally, we tested for multicollinearity in the regression analyses, and all VIF indices were under the value of 5. 

Table 3. Did you use the categorized variables in your regression analyses? I’m wondering because you list being part of the light users group/the under-estimators group/the low checkers group as references.

A: Many thanks for the comment. We decided to remove the categorization of the variables in the main analyses, and we relied only on the continuous log-transformed variables for the analyses. 

Minor issues:

Line 80: Please correct the references and delete the duplicate (10,20,20,21)

Please correct typos, e.g. Internet (line 156), line 455 (64,65,38,66),…

A: Many thanks. We checked the manuscript thoroughly and made the requested corrections. 

Reviewer #3: The paper is concisely written and addresses an important topic – the validity of indicators of problematic smartphone use – assessed via digital trace data. Particularly the addressed sample of adolescents is very interesting for early interventions, and thus, data derived from this population group are valuable. Unfortunately, only a very small subsample (89 out of 1419 students) could be recruited for the current study.

A: Many thanks for the comment. 

• To assess the representativeness of the sample and the results, it would be interesting to compare the reached subsample with the total sample of the Ethica study.

A: This is a good point and we now added information from difference tests showing that the final on subsample did not differ from the original sample. See Samples section lines 239 to 242. 

• The authors argue that self-report data on PSU might not be reliable but validate their digitally traced data also with data from a self-report measure on Smartphone Use (SAS-SV). What about the validity of this instrument? The limitations of this approach should be discussed. Why did you not conduct clinical interviews to assess PSU.

A: Many thanks for the comment. We included the following sentence in the Limitations section to acknowledge the limitation of the SAS-SV scale when it comes to the clinical setting: 

“Finally, although the SAS-SV has been found to predict PSU levels based on the experts' diagnoses, its validation was not carried out for the clinical setting. Hence, the SAS-SV can be used only to detect potentially high-risk groups of PSU, especially in the general population. Future investigations should complement these data with clinical interviews to further validate the use of this scale in assessing symptoms of PSU.”

• The results on the indicator "frequency of checking" are of high interest, I wonder why the authors did not include a measure analoguous to "Perceived duration of smartphone use" to assess "perceived frequency of checking".

A: Many thanks for the comment. We agree that this result would be of great interest and complete the picutre. However, when the larger panel study with annual paper-and-pencil questionnaires was designed, we did not include any questions about the frequency of smartphone use. This should be done in future studies interested in the comparison of self-report and digital trace data. 

6. PLOS authors have the option to publish the peer review history of their article (what does this mean?). If published, this will include your full peer review and any attached files.

Do you want your identity to be public for this peer review? For information about this choice, including consent withdrawal, please see our Privacy Policy.

Reviewer #1: No

Reviewer #2: No

Reviewer #3: No

---

## [Decision Letter · Decision Letter 1]

15 Oct 2021

PONE-D-21-09423R1Duration, frequency, and time distortion: Which is the best predictor of problematic smartphone use in adolescents? A trace data studyPLOS ONE

Dear Dr. Marciano,

Thank you for submitting your manuscript to PLOS ONE. After careful consideration, we feel that it has merit but does not fully meet PLOS ONE’s publication criteria as it currently stands. Therefore, we invite you to submit a revised version of the manuscript that addresses the points raised during the review process.

We look forward to receiving your revised manuscript.

Kind regards,

Michael Kaess, M. D.

Academic Editor

PLOS ONE

Journal Requirements:

Reviewers' comments:

Reviewer's Responses to Questions

**Comments to the Author**

1. If the authors have adequately addressed your comments raised in a previous round of review and you feel that this manuscript is now acceptable for publication, you may indicate that here to bypass the “Comments to the Author” section, enter your conflict of interest statement in the “Confidential to Editor” section, and submit your "Accept" recommendation.

Reviewer #1: All comments have been addressed

Reviewer #2: All comments have been addressed

Reviewer #3: All comments have been addressed

2. Is the manuscript technically sound, and do the data support the conclusions?

Reviewer #1: Yes

Reviewer #2: Yes

Reviewer #3: Yes

3. Has the statistical analysis been performed appropriately and rigorously? 

Reviewer #1: Yes

Reviewer #2: Yes

Reviewer #3: Yes

4. Have the authors made all data underlying the findings in their manuscript fully available?

Reviewer #1: Yes

Reviewer #2: Yes

Reviewer #3: Yes

5. Is the manuscript presented in an intelligible fashion and written in standard English?

Reviewer #1: Yes

Reviewer #2: Yes

Reviewer #3: Yes

6. Review Comments to the Author

Reviewer #1: (No Response)

Reviewer #2: Review on the revised manuscript “Duration, frequency, and time distortion: Which is the best predictor of problematic smartphone use in adolescents? A trace data study”.

The authors used subjective and objective measures of smartphone use at two assessments with a one-year interval and examined the association between the subjective and objective measures, as well as the associations between these measures and the problematic smartphone use score. I have seen the paper before and the authors have been very responsive to the reviewers’ comments. I appreciate the thought and energy that went into this revision. I have only minor comments left:

Could you please clarify the meaning of the reported rs on pages 11 and 12 in the measures section – it is not obvious to the readers

Concerning the descriptions of the questionnaire measures: please keep the descriptions consistent – for some questionnaires (e.g., Children's Social Desirability Short Scale), you report the mean and SD, whereas for others you don’t (e.g., SAS-SV).

Language: the manuscript should be carefully proof-read before publication (maybe by a native speaker)

Reviewer #3: (No Response)

7. PLOS authors have the option to publish the peer review history of their article (what does this mean?). If published, this will include your full peer review and any attached files.

Reviewer #1: No

Reviewer #2: **Yes: **Philip Santangelo

Reviewer #3: No

---

## [Author Response · Author response to Decision Letter 1]

1 Dec 2021

PONE-D-21-09423R1

Duration, frequency, and time distortion: Which is the best predictor of problematic smartphone use in adolescents? A trace data study

PLOS ONE

Dear Dr. Marciano,

Thank you for submitting your manuscript to PLOS ONE. After careful consideration, we feel that it has merit but does not fully meet PLOS ONE’s publication criteria as it currently stands. Therefore, we invite you to submit a revised version of the manuscript that addresses the points raised during the review process.

We look forward to receiving your revised manuscript.

Kind regards,

Michael Kaess, M. D.

Academic Editor

PLOS ONE

Journal Requirements:

Reviewers' comments:

Reviewer's Responses to Questions

Comments to the Author

1. If the authors have adequately addressed your comments raised in a previous round of review and you feel that this manuscript is now acceptable for publication, you may indicate that here to bypass the “Comments to the Author” section, enter your conflict of interest statement in the “Confidential to Editor” section, and submit your "Accept" recommendation.

Reviewer #1: All comments have been addressed

Reviewer #2: All comments have been addressed

Reviewer #3: All comments have been addressed

2. Is the manuscript technically sound, and do the data support the conclusions?

Reviewer #1: Yes

Reviewer #2: Yes

Reviewer #3: Yes

3. Has the statistical analysis been performed appropriately and rigorously? 

Reviewer #1: Yes

Reviewer #2: Yes

Reviewer #3: Yes

4. Have the authors made all data underlying the findings in their manuscript fully available?

Reviewer #1: Yes

Reviewer #2: Yes

Reviewer #3: Yes

5. Is the manuscript presented in an intelligible fashion and written in standard English?

Reviewer #1: Yes

Reviewer #2: Yes

Reviewer #3: Yes

6. Review Comments to the Author

Reviewer #1: (No Response)

Reviewer #2: Review on the revised manuscript “Duration, frequency, and time distortion: Which is the best predictor of problematic smartphone use in adolescents? A trace data study”.

The authors used subjective and objective measures of smartphone use at two assessments with a one-year interval and examined the association between the subjective and objective measures, as well as the associations between these measures and the problematic smartphone use score. I have seen the paper before and the authors have been very responsive to the reviewers’ comments. I appreciate the thought and energy that went into this revision. I have only minor comments left:

Could you please clarify the meaning of the reported rs on pages 11 and 12 in the measures section – it is not obvious to the readers

Authors: We have added a clarification that the r concerns the correlation between weekday and weekend day use.

Concerning the descriptions of the questionnaire measures: please keep the descriptions consistent – for some questionnaires (e.g., Children's Social Desirability Short Scale), you report the mean and SD, whereas for others you don’t (e.g., SAS-SV).

Authors: Many thanks for this observation. We provided the mean and SD for the SAS-SV in the results section, but, for consistency, now added this information also in the measures section.

Language: the manuscript should be carefully proof-read before publication (maybe by a native speaker)

Authors: We performed another careful language check of the manuscript.

Reviewer #3: (No Response)

7. PLOS authors have the option to publish the peer review history of their article (what does this mean?). If published, this will include your full peer review and any attached files.

Do you want your identity to be public for this peer review? For information about this choice, including consent withdrawal, please see our Privacy Policy.

Reviewer #1: No

Reviewer #2: Yes: Philip Santangelo

Reviewer #3: No

---

## [Editor Report · Decision Letter 2]

28 Jan 2022

Duration, frequency, and time distortion: Which is the best predictor of problematic smartphone use in adolescents? A trace data study

PONE-D-21-09423R2

Dear Dr. Marciano,

We’re pleased to inform you that your manuscript has been judged scientifically suitable for publication and will be formally accepted for publication once it meets all outstanding technical requirements.

Kind regards,

Michael Kaess, M. D.

Academic Editor

PLOS ONE
---

## [Editor Report · Acceptance letter]

3 Feb 2022

PONE-D-21-09423R2 

Duration, frequency, and time distortion:
Which is the best predictor of problematic smartphone use in adolescents? A trace data study 

Dear Dr. Marciano:

I'm pleased to inform you that your manuscript has been deemed suitable for publication in PLOS ONE. Congratulations! Your manuscript is now with our production department. 

Kind regards, 

on behalf of

Prof. Dr. Michael Kaess 

Academic Editor

PLOS ONE